# Host and Seasonal Effects on the Abundance of Bean Leaf Beetles (*Ootheca* spp.) (Coleoptera: Chrysomelidae) in Northern Uganda

**DOI:** 10.3390/insects13090848

**Published:** 2022-09-18

**Authors:** Moses Lutaakome, Samuel Kyamanywa, Pamela Paparu, Samuel Olaboro, Charles Halerimana, Stanley Tamusange Nkalubo, Michael Hilary Otim

**Affiliations:** 1Department of Agricultural Production, College of Agricultural and Environmental Sciences, Makerere University, Kampala P.O. Box 7062, Uganda; 2National Crops Resources Research Institute–Namulonge, National Agriculture Research Organization, Kampala P.O. Box 7084, Uganda; 3National Coffee Research Institute, Kituuza, Mukono P.O. Box 185, Uganda

**Keywords:** Chrysomelidae, *Phaseolus vulgaris*, common bean, cowpea beetle, *Ootheca mutabilis*, defoliation, foliage beetle, immature life stages

## Abstract

**Simple Summary:**

Bean leaf beetles (*Ootheca* spp.) (Coleoptera: Chrysomelidae) are endemic in Africa and are pests of cropped legumes (Fabales: Fabaceae). On common beans (*Phaseolus vulgaris* L.), the beetles cause leaf and root damages, leading to losses sometimes exceeding 50% in northern Uganda. Additionally, legumes are often grown in neighboring plots or in rotation season after season. This promotes the multiplication of pests and thus complicate their management. We conducted field trials to assess the seasonal variations in the abundance of the Bean Leaf beetles on common beans, cowpea (*Vigna* sp. Walp) and soybean (*Glycine max* L.), which are major legumes grown in northern Uganda. The beetles were most abundant during the long rainy season of 2018 and on cowpea, but also present during short rainy seasons of 2017 and 2018. Pupae remained in the soil after the harvest of the first season’s crops and were detected during the second season as well. *Ootheca* spp. preferred cowpea for foraging and development. Our study indicates that management strategies should be designed to target the above- and below-ground stages of *Ootheca* spp. in both seasons.

**Abstract:**

Bean leaf beetles (BLBs) (*Ootheca* spp.) are serious legume pests in Uganda and sub-Saharan Africa, but their ecology is not well understood. We planted host plants, viz., common bean, cowpea, and soybean, in an experiment in the hotspot areas of Arua and Lira districts in Northern Uganda in order to assess their influence on the density of adults and immature stages of BLBs in different seasons. Overall, the number of adults, larvae, and pupae were higher in cowpea than common bean and soybean plots. The number of adults were highest in cowpea (29.5 adults/15 plants) in Arua during the long rainy season (2018A). The number of adults did not differ significantly during short rains (season B) in 2017 and 2018. Similarly, in Lira district, the highest number of adult BLBs was in cowpea (4.6 beetles) compared to the common bean (2.7 beetles) and soybean plots, with a peak at four weeks after planting (WAP). During 2018A, larvae of BLBs first appeared at five WAP and seven WAP and peaked at 13 WAP and 11 WAP in Arua and Lira, respectively. The pupae were present in the soil after the harvesting of crops during 2018A, but peaked at seven WAP and eight WAP in 2018B season in Arua and Lira, respectively. The occurrence of below-ground adults in 2018B followed the peak abundance of pupae, although this was delayed until six WAP in Arua compared to Lira. We conclude that cowpea is the most preferred by adults and larvae compared to common bean and soybean. Similarly, the first rain season (2018A) attracted higher abundance and damage than the second rain season. Management of the BLBs should thus take into consideration avoidance of host crop rotation and dealing with the below-ground stages.

## 1. Introduction

Bean leaf beetles *(Ootheca* species) (Coleoptera: Chrysomelidae) (Chevrolat, 1837), hereafter referred to as BLBs, are widely distributed in Sub-Saharan Africa [1]. Thirteen *Ootheca* species have already been described [2], but a few such as *O. bennigseni* Weise, 1900 [3], *O. mutabilis* Sahlberg, 1829 [4], and *O. proteus* Chapius, 1879 [5] are known to be serious agricultural pests. Bean Leaf Beetles are prevalent in Eastern Africa around the Albertine rift valley in Uganda and the Democratic Republic of the Congo [2] as well as western Africa [6]. In Uganda, three species, viz., *O. mutabilis*, *O. proteus*, and *O. orientalis*, were confirmed present [7]. However, only *O. mutabilis* and *O. proteus* exist in the northern region including Lango, Acholi, and the West Nile sub regions, with *O. mutabilis* being the most abundant [5,7]. The reason(s) for the high abundance of *O. mutabilis* in those regions could be among the following: (1) it is more of an environmental generalist than *O. proteus*; (2) continuous or the overlap of rainy seasons in the great northern region; and (3) high diversity of crop and non-crop hosts of bean leaf beetles [7] where the pest would survive in the absence of crop-hosts. The zone grows predominantly legumes (Fabales: Fabaceae), which are the main hosts of *O. mutabilis*. Nevertheless, climate variability and host plant diversity and density are known to influence the population of BLBs [8,9].

Bean leaf beetles have a wide host range including members of the Fabaceae (Fabales) families such as common bean (*Phaseolus vulgaris* L.), cowpea (*Vigna unguiculata* Walp.), soybean (*Glycine max* L.), and pigeon pea (*Cajanus cajan* L.), and Malvaceae (Malvales) such as okra (*Abelmoschus esculentus* L.), *Hibiscus* spp. [10], and *Gossypium* sp. L. [11]. The major crop hosts that influence the life history of BLBs are from the Fabaceae family, which include common bean [8], cowpea [12], and soybean [13]. In Tanzania, [14] reported that the emergence of *O. bennigseni* is influenced by the type of host plants present. Indeed, common bean and cowpea were found to favor the emergence, and perhaps development, of *O. bennigseni* in Tanzania [8]. However, farmers in northern Uganda commonly grow a mixture of common bean, cowpea, and soybean or in rotation with other legumes [15], which promotes the continuity of pest outbreaks. Host plants regulate abundance and population dynamics of insect pests by affecting their developmental duration, fecundity, and life table parameters [16].

*Ootheca* spp. undergo complete metamorphosis and their life cycle is completed in the soil where the juveniles feed on one host to complete development [8]. Mated females oviposit in soil, about 2 cm deep, near the stock of suitable host plants [8]. The eggs hatch into larvae which feed entirely on the root system of their host until pupation [8]. Development durations vary greatly between 60 days on cowpea [17] and 5 months on common bean [8], depending on the weather patterns. After completion of their development, adult beetles emerge from the soil usually in synchrony with rainfall and the presence of host plants [18].

Foraging adults often have a great diversity of hosts from which to choose. After a successful host search, adults skeletonize the leaves and make irregular holes, thus reducing the photosynthetic area and activity. The larvae feed on nodules and secondary roots, resulting in the disruption of movement of nutrients and water. These kinds of damage by adults and larvae lead to seedling death, poor pod filling, and premature senescence [18]. In common bean, grain yield losses ranging from 16.9% to 52.3% have been reported due to damage by adults and subterranean larvae of BLBs in Uganda [5]. In Tanzania, yield losses of 18–32% due to adults of *Ootheca bennigseni* have been reported [3]. Sometimes, damage by BLB completely destroyed the crop [18], and this caused some farmers to abandon growing of the crop [9]. It is important to develop affordable and acceptable control methods. However, the success of a control strategy relies on having adequate knowledge about the pest.

The polyphagous nature of BLBs and the great diversity of host crops also complicate management of the pest in Uganda. In this study, we investigated the seasonal variation in the population of BLBs on common bean, cowpea, and soybean and described the life history of BLBs in two agro-ecological zones in the northern region of Uganda. The knowledge from this study is useful for the proper selection of viable management tools and choice for the mixtures of crops and host plants.

## 2. Materials and Methods

### 2.1. Study Areas

Experimental trials were conducted to monitor the abundance of BLBs and leaf damage at Abi Zonal Agricultural Research and Development Institute (ZARDI) (3°04′ N, 30°56′ E), Ngetta ZARDI (2°17′ N, 32°55′ E) in Arua and Lira district, respectively. Abi ZARDI and Ngetta ZARDI are the Zonal Agricultural Research and Development Institutes of the National Agricultural Research Organization (NARO) located in the West Nile and northern regions in Uganda, respectively. In terms of agroecological delineation, Arua is found in the West Nile Farmlands (WNF), while Lira belongs to the Northern Moist farmlands (NMF) [19]. Arua receives a mean annual rainfall of 1304 mm and annual mean temperature of 23 °C, while Lira district receives a mean annual rainfall and temperature of 1258 mm and 24 °C, respectively. Similarly, the rainfall in Arua is monomodal, while in Lira, it is bimodal, but there is an overlap between the first and second rainy seasons in Lira [19].

### 2.2. Crops Used in the Study

Three host crop plants, viz., common bean (*P. vulgaris*), cowpea (*V. unguiculata*), and soybean (*G. max*), were used in this study. The crops were selected because they have previously been reported as hosts of *Ootheca* spp. [13] and are popular legumes in northern Uganda [15], where *Ootheca* spp. are abundant.

The varieties of host crops used in this study were Kahura for common bean, SeCow2w for cowpea, and MakSoy1N for soybean. Kahura is a red-mottled, medium-seed local variety popular in midwestern (Hoima and Masindi) and northern (Oyam and Lira) Uganda. SeCow2W and MakSoy1N are varieties released by the National Agricultural Research Organization (NARO) and Makerere University (MAK), respectively.

### 2.3. Study Design

An experimental field consisted of 12 plots with the three host crops species planted in a randomized complete block design, which was replicated four times. Two seeds of Kahura per hole with inter- and intra-row spacing of approximately 50 cm and 20 cm, respectively; two seeds of SeCow2W per hole with inter- and intra-row spacing of 50 cm and 30 cm, respectively; and one seed of MakSoy1N per hole with inter- and intra-row spacing of 60 cm and 5 cm, respectively, were planted. A one-meter buffer zone that consisted of three rows of common bean was established around the experimental garden. The experiments were conducted during the 2017B, 2018A, and 2018B seasons, and planting for all the crops was completed at the time when most local farmers sowed their crops at the respective locations. The actual planting dates were the 17th and 18th September 2017 during 2017B; 19th April and 28th March 2018 during 2018A; and 10th and 9th September 2018 during 2018B in Arua and Lira, respectively. The experiment was left under a natural infestation of *Ootheca* spp. and natural enemies.

### 2.4. Assessment of the Above-Ground Abundance and Leaf Damage Caused by Ootheca Species

In order to assess the effects of hosts on the abundance and incidence of BLBs and leaf damage, data were collected weekly starting at 2 weeks after planting (WAP) until 10 WAP, at which time the common bean had reached physiological maturity. To assess the influence of host growth stages, data on abundance and incidence of BLBs and leaf damage were collected weekly starting at 2 WAP until each crop reached maturity, corresponding to 10 WAP for common bean and 14 WAP for both cowpea and soybean. Four growth stages, namely, seedling, vegetative with branching, flowering and pod formation, and pod maturation, were adopted [20] (Appendix A). At each instance of data collection, 10 plants in 2017B and 15 plants in 2018A and 2018B were selected according to [9], following a zig-zag pattern within the plot. The number of sampled plants was increased from 10 to 15 because of the low population of BLB encountered during the 2017B season, and sampling for the abundance of BLB on all host plants was stopped at 10 WAP due to the very low population of BLBs during the 2018B season. For young plants, adult BLBs were directly counted, but after the start of branching (V4 stage and older stages), adults were collected onto a white paper placed below the canopy by shaking the plant. At this stage, plants were too bushy to allow for the direct counting of beetles. Shaking the plants dislodged beetles that were hiding in the canopy. In order to minimize flights of adults between plots and plants, an assessment for the abundance of adult BLB was performed between 0800 h and 1100 h. During the sampling period, the dew on the canopy had also dried, yet the adults exhibited minimum activity. The dew would otherwise prevent adults from migrating to the canopy, which would affect estimation of their abundance. The few beetles that flew away from the sampled plants were also counted, as long as they never left the plot. The beetles that left the plot for the next were not counted in order to avoid double enumeration. The *Ootheca* spp. were identified based on keys provided by [2] and [1]. Leaf damage severity was scored using a visual scale of 0 to 5 where 0 = no damage; 1 = 1 to 5%; 2 = 6 to 25%; 3 = 26 to 50%; 4 = 51 to 75%; and 5 = 76 to 100% damage [3]. All the leaves on the entire plant were assessed and awarded a single average score.

### 2.5. Assessment of Below-Ground Life Stages of Ootheca spp.

Data on abundance on below-ground *Ootheca* spp. life stages were collected weekly in two randomly marked quadrats per plot, each measuring 1.0 m by 1.0 m. The shoots of plants in each quadrat were cut one by one at ground level and removed from the quadrat area. The remaining stems were scooped with a hand hoe to a depth of approximately 20 cm, while keeping the soil on their root systems intact. The intact soil blocks were then placed on a black polythene mat and carefully broken to expose any life stages of BLB associated with the root system (referred to as ‘on-plant’). The soil in the inter-row space covered in the quadrat was also scooped to the same depth, placed on the same mat, and examined as above (referred to ‘further than 20 cm’) and searched. The recovered BLB life stages were counted and recorded according to their location in the soil. The interval between sampling during offseason sampling was increased to three weeks because the soil became so hard that it was difficult to break the lumps. This period occurred at 11 WAP onwards, including the time when crops were already harvested (offseason).

### 2.6. Data Analysis

Since each growth stage spanned more than one WAP, an average count for each plot was obtained before analysis [6]. Plants with BLBs were calculated as the percentage of plants with BLBs out of the total plants sampled in a plot. Statistical analyses and means for the abundance of BLBs, leaf damage, and percent plants with BLBs were calculated on samples collected from 2 to 10 WAP.

Data analysis was performed in GenStat version 14th edition for windows (https://genstat.kb.vsni.co.uk) (accessed on 1 May 2021) and Excel (2013) version 15 produced by Microsoft corporations, Redmond, WA, USA. Prior to the analysis, data were subjected to the Shapiro–Wilk test for normality in Genstat, which found that the significance value was lower than the threshold 0.05, indicating that none of the transformed data were very far from being drawn from a normally distributed population (W = 0.3046, Probability: <0.001). In order to stabilize the variance, data on BLB count and percent plants with BLBs were Log (x + 4) transformed. The same transformation was applied to the counts for larvae, pupae, and below-ground adults. To evaluate the effects of host plants, cropping seasons, and locations on the abundance of, leaf damage due to, and incidence of *Ootheca* spp., we conducted a repeated measures analysis of variance (RMANOVA) with host plant x season x district as the treatment structure and replications as the blocking structure [21]. To determine the effects of host growth stages and interaction with host plant types, we conducted a two-way ANOVA on the abundance of BLBs with host plant x growth stage as treatments and season x replication as the blocking structure. We also conducted a Pearson correlation to determine abundance’s relationship with rainfall and temperature. Whenever significant differences were detected, the multiple and district means were separated using Tukey’s honestly significant difference (HSD) and Fisher’s *t*-test, respectively.

## 3. Results

### 3.1. Influence of Season and Host Plants on the Abundance, Distribution, and Leaf Damage by Ootheca Species

The main effects of season, host plant species, and location significantly influenced the abundance, leaf damage, and proportion of plants infested with bean leaf beetles (Appendix A). The abundance of bean leaf beetles was significantly (df = 2, 51, F = 816.83, *p* < 0.001) higher in 2018A than in both 2017B and 2018B (Table 1). Leaf damage was also significantly (df = 2, 51, F = 194.26, *p* < 0.001) higher in 2018A than 2017B, which in turn was higher than 2018B (Table 1). Similarly, the proportion of plants infested with bean leaf beetles was significantly (df = 2, 51, F = 1224.69, *p* < 0.001) higher in 2018A than 2017B, which in turn was higher than 2018B (Table 1).

There were significantly more bean leaf beetles (df = 1, 51, F = 546.62, *p* < 0.001), higher leaf damage (df = 1, 51, F = 194.21, *p* < 0.001), and a higher number of plants infested with bean leaf beetles (df = 1, 51, F = 725.31, *p* < 0.001) in Arua than in Lira (Table 1). 

The abundance of bean leaf beetles was significantly (df = 2, 51, F = 75.14, *p* < 0.001) higher in cowpea than in common bean plots, which also had significantly higher beetles than soybean (Table 1). Leaf damage (df = 2, 51, F = 254.78, *p* < 0.001) and the proportion of plants infested with bean leaf beetles (df = 2, 51, F = 85.10, *p* < 0.001) were similar and significantly higher in both cowpea and common bean than in soybean plots (Table 1). 

### 3.2. Influence of Host Plant Phenological Stages on the Abundance of Ootheca Species

The number of BLBs was highest (df = 3, 143, F = 9.82, *p* < 0.001) during the vegetative with branching stage and lowest during the pod maturation stage in Arua district (Figure 1). However, cowpea plots harbored most beetles during flowering and pod formation, though they did not differ from those during vegetative with branching (Figure 1). In Lira, the number of adult beetles was highest (df = 3, 143, F = 9.29, *p* < 0.001) at the vegetative with branching stage, and the pattern of crop infestation by beetles through the growth stages was the same as in Arua. The lowest population of adult beetles was recorded at the pod maturation stage. There was no significant interaction between the host plant and growth stages (*p* < 0.05) at either location.

### 3.3. Seasonal Abundance of BLBs, Leaf Damage, and Incidence Fir Common Bean, Cowpea and Soybean

There was a significant interaction between season and host plants for the number of BLBs (df = 4, 51, F = 53.54, *p* < 0.001), leaf damage (df = 4, 51, F = 10.60, *p* < 0.001) and the proportion of plants with BLBs (df = 4, 51, F = 28.22, *p* > 0.001). Adult BLBs occurred in the first (A) and second (B) rainy seasons in the two districts, but with more BLBs recorded in the first than in the second rainy season (Figure 2). The population of BLBs in the second rainy seasons (2017B and 2018B) was erratic and considerably lower than in 2018A.

**Abundance of BLBs during 2017B and 2018B seasons**: Adult BLBs were observed at three WAP during 2017B on cowpea in Lira. No adult BLBs were observed on common bean and soybean throughout the 2017B season at the same location. However, in Arua, the number of adult BLBs peaked at six and eight WAP on cowpea and at eight WAP on common bean during the 2017B season. During 2018B, adult BLBs were observed on common bean at two WAP to five WAP, on Cowpea from four WAP to seven WAP, and on Soybean at six WAP. No adult BLB was observed on any of the host crops in Arua during 2018B.

**Abundance of adult BLBs during 2018A season:** The adult BLBs were observed in the fields soon after the emergence of the crops. In Arua, the number of adult BLBs peaked at eight WAP on cowpea, five WAP on common bean, and four and eight WAP on soybean. The rainfall was highest at four WAP and gradually decreased thereafter (Appendix A); however, a non-significant negative linear correlation was observed for rainfall (df = 72, r = −0.006. *p* > 0.05) and temperature (df = 37, r = −0.254, *p* > 0.05). In the same season, the number of adult BLBs on all crops peaked at four WAP in Lira. The rainfall was highest at the beginning of the season in Arua with a peak at four WAP (95.4 mm), but was quite even in Lira during the 2018A season with a peak at two WAP (Appendix A). The amount of rainfall received throughout the study period in Arua (2450 mm) was roughly similar to that in Lira (2591 mm), with only 1359 mm and 1490 mm received during 2018, respectively (Appendix A).

### 3.4. Influence of Host Plants on the Abundance and Distribution of Life Stages of Ootheca spp. (BLB) below Ground

The abundance and distribution of immature life stages of BLBs among host plants is presented in Table 2. The number of larvae (df = 2, 51, F = 23.23, *p* < 0.001), pupae (df = 2, 51, F = 13.24, *p* < 0.001), and below-ground adults (df = 8, 51, F = 17.88, *p* < 0.001) were significantly different among hostplant species (Appendix A). The mean numbers of larvae, pupae, and teneral adults recovered were highest in cowpea, followed by common bean, and least on soybean plots. The number of larvae recovered in common bean plots was not statistically (Table 2) different from that on soybean within 20 cm around and away from plants. The number of pupae recovered, just like larvae, were highest on cowpea in soils 20 cm away from plant; however, there was no difference within 20 cm diameter around plants (Table 2). Similarly, the mean number for below-ground adults was higher on cowpea than on common bean and soybean, but the number of adults recovered in cowpea was not different from that on common bean (Table 2).

### 3.5. Seasonal Variation in Abundance of Below-Ground Immature Life Stages of BLB across Arua and Lira Districts

The number of larvae (df = 2, 51, F = 105.58, *p* < 0.001), pupae (df = 2, 51, F = 24.36, *p* < 0.001), and below-ground adults (df = 2, 51, F = 35.18, *p* < 0.001) were different among three seasons (Appendix A). There were also significant interactions between season and location in the abundance of larvae (df = 2, 51, F = 31.95, *p* < 0.001) and below-ground adults (df = 2, 51, F = 10.24, *p* < 0.05), but not for pupae (df = 2, 51, F = 1.77, *p* > 0.10) (Appendix A). The abundance of pupae was higher during 2018B than in 2017B and 2018A in both locations (Table 3). The abundance of below-ground adults among seasons was only significantly different in Arua (Table 3), where the highest number was recorded in 2018A rather than 2017B and 2018B seasons. In Lira district, the mean below-ground adults per plot was higher during 2018B, but not statistically different from that during 2017B and 2018A (Table 3).

The variations in the abundance of BLB life stages across weeks after planting (WAP) during the host crop growth cycle and post-harvest are presented in Figure 3. The number of larvae, pupae, and below-ground adults of BLBs peaked at different times in the three different seasons.

The mean larvae of BLB per plot was highest at 13 WAP in Arua and 11 WAP in Lira during 2018A throughout the study period. The 13 and 11 WAP corresponded to the post-harvest and pod maturation stage for common bean, respectively. The mean larvae of BLB per plot steadily decreased thereafter towards the end of the season at both locations. A low number of larvae were observed at one to two WAP in Arua district; however, no larvae were observed in Lira district during 2017B. Similar trends were observed during the 2018B season except that the high abundance of larvae was observed at the beginning of the season, but later reduced gradually.

During 2018A, pupae were first recorded at 14 WAP in both Arua and Lira districts. The abundance of pupae increased thereafter through the end of the season. During 2017B, the abundance of pupae was highest at the start of the season and gradually reduced through 12 and 7 WAP in Arua and Lira, respectively. This was a similar trend as in 2018B; however, the abundance of pupae slightly increased at the beginning of the season to the peak at seven and eight WAP in Arua and Lira, respectively. The abundance of pupae reduced gradually thereafter through 10 and 14 WAP in Arua and Lira, respectively.

The abundance of below-ground adults was highest at the start of the 2018A season in the Arua and Lira districts, and reduced steadily until nine WAP. During 2017B, mean teneral adults gradually increased, starting at four WAP in the Arua and Lira districts. This trend was similar for 2018B at both locations, except that the abundance of teneral adults was higher, and they were first observed at six WAP in Arua district. A majority of the teneral adults recovered during 2018A were active feeders, whereas those recovered during 2017B and 2018B did not feed when put in cages containing fresh bean plants.

## 4. Discussion

This study investigated the effects of location, season, host plant species, and their phenology on the temporal distribution and abundance of the life stages of *Ootheca* spp. The results showed that cowpea was the most preferred host of BLB life stages, followed by common bean and soybean. Furthermore, the abundance of BLB life stages was influenced by the location, season, and phenology of the host crops. In addition, similar to [7], we report that BLBs occurred in the second season, albeit in lower numbers when compared with the first season.

The higher abundance of adult BLBs on cowpea than on common bean and soybean indicates that cowpea is the most preferred host, followed by common bean. Consequently, more eggs, reflected by higher larval counts, appeared to have been laid in cowpea plots resulting in a higher population of BLBs in different life stages. The reasons for this particular order of preference for the different hosts are not known. However, insects are known to be attracted by specific volatiles that are emitted by their host plants [22] and prefer more nutritious plants for progeny development [23]. The higher preference for cowpea could be due to the fact that the regions that hosted the experiment sites are dominated by the cowpea leaf beetle (*O. mutabilis*) [7], which is predominantly a pest of cowpea [6]. These results on preference agree with earlier reports by [9], where cowpea-bean attracted higher numbers of *Ootheca* species than sole bean plots in Apac district in Uganda. The preference of cowpea over common bean suggests that the crop can be used as a trap against bean leaf beetles in common bean fields. In this case, common bean would be the major crop, and cowpea the minor one. However, further research is necessary to identify the factors for preferential attraction to cowpea since these were not established during this study.

The abundance of BLBs was highest at the vegetative stage, except for cowpea, where population peaked at the flowering stage. The abundance of BLBs on cowpea increased even after the onset of flowering. The reasons for the high abundance of BLBs at the vegetative stages are because *Ootheca* spp. emerge at the onset of rain, when the host crops germinated, thus coinciding with the vegetative stage, and this was consistent with the 2018A season and supported by [8]. Additionally, the leaves were tender and perhaps more palatable at the vegetative stage, which non-preferentially attracted foraging adults towards host crops. However, due to the reduction in the abundances of adults on common bean and soybean throughout the flowering stage, it can be suggested that adult BLBs migrated from plots to cowpea plots, which is supported by [9]. *Ootheca* spp. also migrate from nearby fallowed fields soon after the host emerges [24]. Cowpea plants also continually put on fresh leaves through pod setting, which attracted more adult BLBs. On the other hand, the leaves of common bean and soybean were coarse and may have become less palatable to BLBs after the onset of flowering. A study by [6] supports the argument that *O. mutabilis* would migrate from older fields to vigorously growing ones, and older leaves of cowpea became lignified, which deterred feeding by the insect herbivores, hence the decrease in the abundance of *O. mutabilis*. Due to this reason, we suggest that whenever manipulating the planting time is used as a control tool for the management of BLB, it should be synchronized with fields over a wide area. This way, defoliating adults would be evenly distributed, and no field would act as source or carry the burden of defoliation. On the other hand, it can be hypothesized that female BLBs are actively searching for and ovipositing on young hosts before flowering. This is supported by the high spike of larval instars immediately after the onset of flowering on cowpea, common bean, and soybean. However, due to challenges associated with the collection of eggs in the field, an area-wide survey for egg clusters in the field was not conducted.

Despite the very low abundance of BLBs in 2017B, leaf damage was considerably high in this season. Other defoliators from two genera, viz., *Nisotra* and *Medythia*, were detected during the 2017B season, which could have contributed to some of this damage. Similar observations were reported by [7], whereby *Nisotra* spp., *Lema* spp., and *Diacantha* spp. caused holes on leaves similar to those by BLBs.

The abundance of BLBs differed considerably between districts (locations). The number of adult BLBs was seventeen times higher in Arua than that in Lira district. Similar results were obtained for the abundance of immature stages including teneral below-ground adults. Some of the major factors that may influence insect distribution and abundance are seasonal variations in the environmental conditions and host diversity [25,26]. The overall total rainfall received may not have had a direct impact, but rather indirectly affected distribution patterns and the abundance of adult BLBs and their life stages. The high rainfall received at three WAP influenced the emergence of BLBs from the soil, evidenced by the drop in the numbers of below-ground adults, and it may have promoted plant vigour, which may have attracted foragers from neighboring fields in Arua during the 2018A season. This reasoning is supported by [8], where BLBs emerged at the start of seasonal rains in Tanzania. On the contrary, the abundance of BLBs may have been affected negatively by constant rainfall distribution during the same season (2018A), since it rained at least once every 7 days (Appendix A) during the study period in Lira district. During this time, the adult BLBs could be dislodged from the plant canopy and drowned in small puddles on the ground.

*Ootheca* spp. were present in the second season, although in lower numbers than in the first season. This phenomenon is more pronounced in Arua district, which may be due to the unimodal nature of rainfall [19]. However, previous reports by [9] only indicated the occurrence of BLBs in the first season. The adults recorded during the second season are most likely the same generation as those recorded during the first season, as these were not reflected in the teneral adults or pupae and larvae recovered during the start of the second season. The observation from the present study indicated one annual generation of BLB whereby the adults observed during the 2018A season formed towards the end of 2017B. This implies that adults that form during the second rain season would emerge from the soil during the first season of the following year. Adult BLBs are formed in the soil during the end of season B and emerge at the start of season A of each year, where we suggested there would be a diapause period of adulthood before emergence. The adults recovered from soil during season B had pale colors, hardly walked, and never fed. However, these findings are contradicted by [17], where *O. mutabilis* underwent facultative diapause. The first generation (without dormancy) occurred before cowpea matured, but the second generation diapaused after crop maturity and emerged the following year [17]. This study, however, did not establish the cues for breaking dormancy in BLBs.

The findings from this study have implications towards the management of *Ootheca* species in tropical region in Africa. Foliar and root damage by adults and larvae, respectively, are more important in the first season than in the second. Nonetheless, larvae and teneral adults exist in the soil in the second season. The larvae and teneral adults would become a problem when mature during the following year, which implies that as one manages the foraging adults, emphasis must also be put on the other life stages (larvae and pupae), but in different seasons. Soil drenches with economically viable pesticides can help to reduce larval damage on the root system and boost pod setting in the first rainy season. Foraging adults are controlled during the same season. Similarly, post-season ploughing will help to expose the soil-dwelling pupae and teneral adults to predators and harsh environment conditions such as desiccation from the sun’s heat at the end of first season, thus reducing the populations of the beetles in the following season [27]. Ploughing before planting for the second season should be useful as well post-season. Similarly, intercropping cowpea or planting beans near cowpea fields should be avoided, except if used as trap crops to divert populations and damage.

## 5. Conclusions

Overall, cowpea attracted the highest number of adults and larvae of BLB across the two locations during all seasons. Similarly, the population of adults and larvae of BLB occurred in the first and second season, but higher during the former. The abundance of adult BLBs was highest at the vegetative with branching stage for common bean and soybean, and the flowering and pod formation stage for cowpea. Growers of the host crops such as common bean and cowpea should invest in strategies for BLB control in the two seasons. Control of BLB in the first season would protect the crops from defoliating adults and damage from subterranean larvae, but also mitigate outbreaks during the subsequent season and year.

## Figures and Tables

**Figure 1 insects-13-00848-f001:**
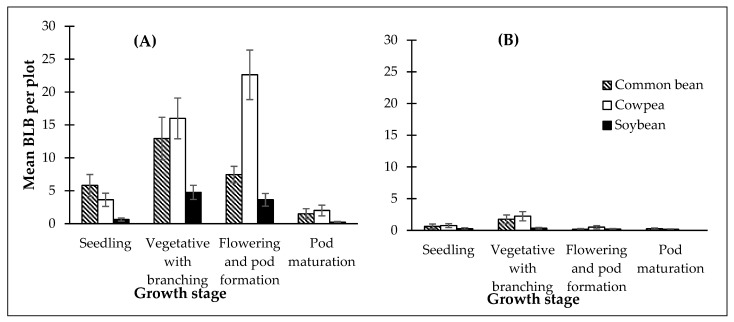
Abundance of adult *Ootheca* spp. at different growth stages of common bean, cowpea, and soybean in Arua (**A**) and Lira (**B**), averaged over seasons. Error bars represent standard errors.

**Figure 2 insects-13-00848-f002:**
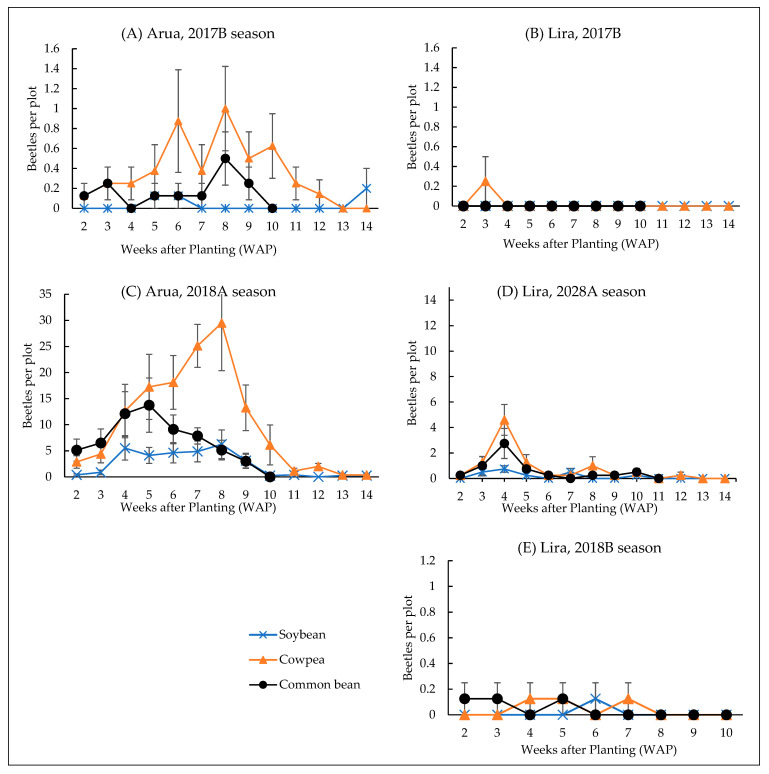
Abundance of adult *Ootheca* spp. on common bean, cowpea, and soybean in Arua and Lira district during the short rain season in 2017 (**A**,**B**), long rain (**C**,**D**) and short rain (**E**) seasons in 2018. The common bean growth stages of seedling, vegetative with branching, flowering and pod formation, and pod maturation correspond for two to three, four to five, six to eight, and nine to eleven weeks after planting, respectively. There is no graph for Arua, 2018B season because no beetles were recorded at the location during that season.

**Figure 3 insects-13-00848-f003:**
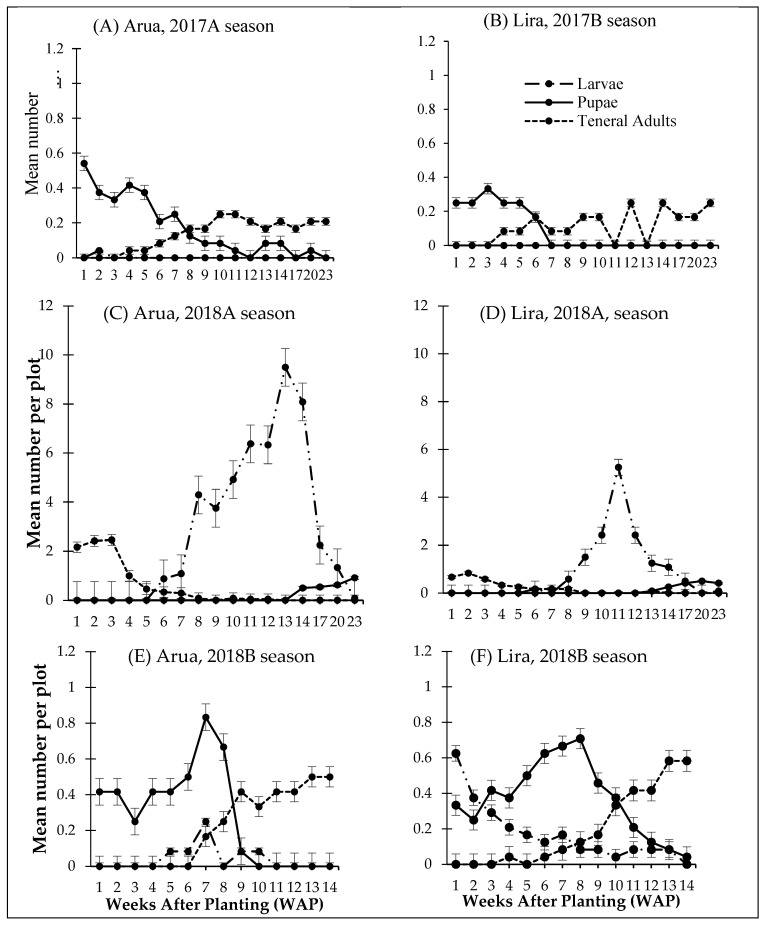
Abundance of below-ground *Ootheca* spp. life stages across three seasons in the Arua and Lira districts. Left column (**A**,**C**,**E**)—Arua district; right column (**B**,**D**,**F**)—Lira district. Top row: 2017B, middle row: 2018A, and bottom row: 2018B season.

**Table 1 insects-13-00848-t001:** Abundance of adult *Ootheca* spp. (mean ± s.e), leaf damage and incidence across seasons, districts, and host plants.

Factor	Number of BLBs per Plot	Leaf Damage	Plants with BLB (%)
Season effects			
2017B	0.16 ± 0.03 ^a^	1.33 ± 0.03 ^b^	1.5 ± 0.27 ^b^
2018A	5.72 ± 0.58 ^b^	1.71 ± 0.04 ^c^	21.5 ± 1.71 ^c^
2018B	0.02 ± 0.01 ^a^	1.11 ± 0.05 ^a^	0.1 ± 0.05 ^a^
District (Location) effects			
Arua	3.39 ± 0.37 ^b^	1.57 ± 0.03 ^b^	13.0 ± 1.12 ^b^
Lira	0.19 ± 0.03 ^a^	1.15 ± 0.04 ^a^	1.2 ± 0.21 ^a^
Host plant effects			
Common bean	1.66 ± 0.28 ^b^	1.56 ± 0.05 ^b^	7.5 ± 1.14 ^b^
Cowpea	3.45 ± 0.53 ^c^	1.63 ± 0.04 ^b^	11.5 ± 1.4 ^b^
Soybean	0.78 ± 0.15 ^a^	0.96 ± 0.03 ^a^	4.1 ± 0.76 ^a^

Note: For each factor, means in the same column followed by different letter superscripts are significantly different (*p* ≤ 0.05).

**Table 2 insects-13-00848-t002:** Mean number of larvae, pupae, and below-ground adults of *Ootheca* spp. recovered within and further than 20 cm around the crop in plots of different host plants.

BLB Life Stage A	Host Plant	Mean Number Recovered per Plot *	
		Within 20 cm Diameter around Plant	Further than 20 cm from Plant
Larvae	Common bean	0.69 ± 0.258 ^a^	1.25 ± 0.467 ^a^
	Cowpea	3.17 ± 0.922 ^b^	5.55 ± 1.293 ^b^
	Soybean	0.28 ± 0.117 ^a^	0.22 ± 0.077 ^a^
Pupae	Common bean	0.16 ± 0.052 ^a^	0.60 ± 0.080 ^b^
	Cowpea	0.21 ± 0.070 ^a^	1.22 ± 0.153 ^c^
	Soybean	0.07 ± 0.029 ^a^	0.10 ± 0.037 ^a^
Below-ground adults	Common bean	0.64 ± 0.282 ^a^	0.62 ± 0.138 ^b^
	Cowpea	0.69 ± 0.250 ^a^	1.01 ± 0.163 ^b^
	Soybean	0.16 ± 0.073 ^a^	0.08 ± 0.032 ^a^

* Number of life stages represent the total number of life stages recovered from eight quadrats corresponding to four plots per host plant per one WAP. Means of life stage in the same column followed by different letters are significant (*p* < 0.05).

**Table 3 insects-13-00848-t003:** Mean number of larvae, pupae, and teneral adults of *Ootheca* spp. recovered in the soil during 2017B, 2018A, and 2018B seasons in Arua and Lira districts, 2017A to 2018B.

BLB Life Stage	Season	Mean Number per Plot (mean ± s.e.m) ^¥^	
		Arua	Lira
Larvae	2017B	0.002 ± 0.245 ^a^	0
	2018A	2.960 ± 0.249 ^b^	0.897 ± 0.096 ^b^
	2018B	0.042 ± 0.382 ^a^	0.173 ± 0.075 ^a^
Pupae	2017B	0.18 ± 0.028 ^a^	0.09 ± 0.042 ^a^
	2018A	0.13 ± 0.028 ^a^	0.10 ± 0.042 ^a^
	2018B	0.29 ± 0.043 ^b^	0.37 ± 0.033 ^b^
Below-ground adults	2017B	0.14 ± 0.058 ^a^	0.11 ± 0.036 ^a^
	2018A	0.57 ± 0.059 ^b^	0.19 ± 0.036 ^a^
	2018B	0.21 ± 0.090 ^a^	0.20 ± 0.028 ^a^

^¥^ Means of life stages in the same column followed by different letter superscript are significant (*p* < 0.05).

## Data Availability

All data are provided in the main body of the published article and Appendix A.

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
