# Peer review of "Host and Seasonal Effects on the Abundance of Bean Leaf Beetles (Ootheca spp.) (Coleoptera: Chrysomelidae) in Northern Uganda"

_insects, 2022, doi:10.3390/insects13090848_

Round 1

Reviewer 1 Report

Review of MS - Insects-2022—“Host and Seasonal effects on the abundance of Bean Leaf Beetles (Ootheca spp.)  (Coleoptera: Chrysomelidae) in northern Uganda”

The MS addresses a topic concerning on phenology and host preference of herbivore Ootheca spp. This group represents a serious pest that causes considerable damage to legumes in large areas of tropical Africa. In consideration of the need to control the swarms of these species, it is necessary to know in greater detail the bio-ecology of these herbivores.

Several considerations on the MS are necessary to carry out a rearrangement of the same from the summary.

-It is not clear in the MS whether the identification of the species or different species was carried out. If several species are found in relation to the sowing time, it is essential to identify the species.

-In the MS it is necessary to specify if phytoiatric interventions have been made to control other parasites or fungal diseases.

Line 15. It is advisable to remove “often reaching 52.3%”.

Line 21 and line 31. The abbreviations entered (example 2018A) do not make the sentences clear to the reader who can only guess. It is requested to modify this outline and to reformulate the summaries.

Line 69-71. This sentence gives the impression that it is only adults who cause damage to legumes. Instead, the presence of the larvae is not mentioned.

Line 113. It is not necessary to write RCBD with the first letter capitalized.

Line 127. WAP is inserted in the text for the first time, so write the name in exstenso as well.

Line 130. Report everything in writing what has been done and what has not been done [9]. It is not always easy to find the bibliographic reference of material not present in international data banks.

Line 141-143. Specify if the leaf damage estimate was made on randomly picked leaves or if the leaves of a certain leaf stage were chosen. I also find it risky to specify 5 damage classes and identify the correct class. The estimate was visual or you used something to measure the eroded surface.

Line 159. Data Analysis. The paragraph needs to be specified better. The transformation of the data should be justified only if the data requires it so you have to do some tests. Why was multivariate analysis chosen? Furthermore, categorical variables need to be better specified.

 Figure 1 does not show the monitoring year. Were the monitored adults grouped for the two-year period?

 Figure 2 relates the climatic variables with the abundance of leaf beetles but in consideration of the double y-axes the visualization becomes complicated. If you do not want to change the size of the axes to make the abundance data more readable, indicate in the caption what the color indicates in the caption.

 Line 320-325. It is difficult to understand what the authors wrote. I wonder if it was not possible to distinguish the other defoliators for 2017B? During the sampling it was not possible to collect and identify them. This aspect is important as the host preference of other herbivores is not known.

Line 339-340. Although the hypotheses may be plausible such as the mechanical effect of rain, other more important effects must also be considered, especially for herbivores that spend part of their cycle in the soil. Among these effects, for example, there could be the action of entomopathogenic terrestrial fungi that reduce the number of emerged ones.

Line 364-365. Specify in accordance with crops practices whether soil tillage can have a positive effect on the reduction of pest.

Reviewer 2 Report

1. I think the Introduction is short and it needs to be improved. The authors did not specify many necessary references for the introduction. Therefore, the references in the manuscript is very small.

2. Line 289. The phrase ".... or prefer more nutritious plants for progeny development". Specify the publications.

3. Line 300-306. What evidence can you provide for these assumptions?

4. Line 338-340. Proof is needed.

5. Is it possible to use the sowing of other plants after legumes as an agrotechnological technique?

Round 2

Reviewer 1 Report

The authors modified and added the parts suggested in the first revision. The MS now is improved. Few are typos.

Author Response

The typos suggested have been corrected by subjecting the manuscript to a spelling and grammar check in MS word and proof reading by Authors.